# Development of Warm In-Place Recycling Technique as an Eco-Friendly Asphalt Rehabilitation Method

Byungkyu Moon [1], Ashkan Bozorgzad [1], Hosin (David) Lee [1,*], Soo-Ahn Kwon [2], Kyu-Dong Jeong [2] and Nam-Joon Cho [3]

[1] Iowa Technology Institute, University of Iowa, Iowa City, IA 52242, USA; byungkyu-moon@uiowa.edu (B.M.); ashkan-bozorgzad@uiowa.edu (A.B.)

[2] Korea Institute of Civil Engineering and Building Technology (KICT), Highway and Transportation Research Institute, Goyang-si 10223, Korea; sakwon@kict.re.kr (S.-A.K.); kdjeong@kict.re.kr (K.-D.J.)

[3] School of Energy, Korea University of Technology and Education, Materials & Chemical Engineering, Cheonan-si 31253, Korea; njuncho@koreatech.ac.kr

* Correspondence: hosin-lee@uiowa.edu; Tel.: +01-801-512-4202

**Abstract:** Cold In-place Recycling (CIR) has been widely used in the world since it is easy to apply it in the field at a low cost. However, it is not normally used as a surface layer as a result of its inconsistent quality due to an excessive amount of fine aggregates pulverized during the milling process. Hot In-place Recycling (HIR) can retain the original shape of the aggregates, but it often produces a large amount of Volatile Organic Compounds (VOCs). Therefore, a third in-place recycling technique is introduced in this paper: Warm In-place Recycling (WIR). The WIR technique overcomes the limitations of both CIR and HIR techniques by lowering a heating temperature while adding a Tetraethylenepentamine (TEPA)/Soybean/SBS additive. To identify the effect of the additive on the RTFO-aged binder, viscosity and dynamic modulus values were measured at different temperatures. Based on Hamburg Wheel Tracking (HWT) and Disc-Shaped Compact Tension (DCT) tests, the additive improved the moisture susceptibility and low temperature cracking resistance. The indirect infrared heating equipment reduced the emission by lowering the pavement surface heating temperature by 20 °C from 140 to 120 °C. Compared with the heating at 140 °C, the LPG usage for heating at 140 °C was lowered by 21%. The proposed WIR equipment with an additive would revolutionize the in-place recycling practices.

**Keywords:** warm in-place recycling; eco-friendly asphalt rehabilitation method; emission controlled heating equipment; tetraethylenepentamine (TEPA); soybean; SBS polymer

## 1. Introduction

The asphalt recycling is one of the most effective ways to rehabilitate asphalt pavements while preserving both construction materials and the environment [1]. A significant effort has been made to increase the content of Reclaimed Asphalt Pavement (RAP) in the Hot Mix Asphalt (HMA) plant using fractionation, soft asphalt binder, and rejuvenators [2–4]. The use of RAP in the US increased by 8.5% in 2019 with an average of 21.1%. Thirty-one US states utilized 20% of RAP or greater in 2019, which include three US states of Ohio, Florida, and Maryland with 32%, 31%, and 30%, respectively [5].

To increase the RAP content, various rejuvenators along with the preheated RAP have been tried. Mixtures containing the preheated RAP binder showed an increase in brittleness and cohesion of the material with the lowest deformability due to the stiffening process linked to the blending occurring between aged and virgin binders [6]. The high RAP content with rejuvenators improved both the rutting resistance and the low temperature performance [7]. Both the commercial rejuvenator and wasted engine oil increased the mixtures' rutting resistance [8]. Bio-based rejuvenators improved the mechanical performances

of RAP binder at low temperatures and decreased the fracture toughness temperature to a level similar to the virgin binder [9].

The interaction between the aged RAP binder and the virgin binder resulted in the blended binder with increased stiffness and elasticity [10]. Based on both lab tests and trial test sections, a rejuvenator allowed the compaction temperature of high RAP mixes to be lowered by 20 °C from the standard temperature used for the hot mix [11]. The workability temperature was lowered using paraffin wax, which can preserve the elastic characteristics of the bitumen and slow the oxidative aging processes during mixing [12]. Rejuvenators can lower the viscosity and stiffness of RAP binder resulting in the increased stress relaxation capacity and fracture energy [13]. However, the reduction of stiffness using a rejuvenator or a soft binder could increase the rutting potential [14]. Among the existing rehabilitation methods of asphalt pavement, the in-place recycling techniques are considered most economical and sustainable with 100% RAP mixtures. In-place recycling techniques can be categorized into the following two types: Cold In-place Recycling (CIR) and Hot In-place Recycling (HIR) methods. Based on the recent survey of 11 US states in the North Central Asphalt User/Producer Group (NCAUPG) region, CIR was 91% whereas HIR was 9% [15]. The CIR produces an irregular pavement surface condition with an inadequate strength due to excessive amounts of fine aggregates pulverized during the milling process [16]. Therefore, to provide a better performing surface layer, an HMA overlay is normally applied on top of the CIR layer [17]. Extensive research has been performed on mix design, curing, and field performance of CIR in the US [18–23]. Reduced greenhouse gas due to the no hauling distance was also reported as an advantage of a CIR [24].

The HIR aims at restoring the surface defects by milling and placing back a thin surface layer after applying in situ heat and, therefore, its application has been limited to pavements with a relatively good condition [25]. The HIR mix can meet the Superpave volumetric requirement since the change in the fine aggregate angularity was very minor due to very little degradation of the fine aggregate during the heated surface milling process [26]. The optimum preheating temperature of HIR is difficult to achieve since the heating levels are affected by environmental conditions [27]. Due to the excessive heating of the pavement surface, however, the HIR process often produces smoke with a large amount of Volatile Organic Compounds (VOCs) [28].

The main objective of this research is to develop a third in-place recycling technique: Warm In-place Recycling (WIR). The proposed WIR technique can overcome the limitations of HIR technique by lowering a heating temperature using indirect infrared heating equipment while adding a warm mix asphalt additive. First, the newly developed emission-controlled heating equipment is discussed. Second, two WIR additives for 100% RAP materials are evaluated for optimum dosage rates based on the viscosity and Dynamic Shear Rheometer tests (DSR). Third, the moisture susceptibility of WIR mixtures is evaluated using the Hamburg Wheel Track (HWT) testing equipment. Fourth, the low-temperature cracking resistance of WMA mixtures is determined using the Disc-Shaped Compact Tension (DCT) testing equipment. The proposed WIR solution allows reducing the emission of harmful compounds into the atmosphere, while obtaining good properties of the pavement layer.

## 2. Emission Controlled Warm In-Place Recycling Equipment

A new WIR equipment was developed for recycling asphalt pavements with significantly reduced emission amounts. The purpose of this WIR technology is to recycle old asphalt pavements in the field while reducing the amount of carbon dioxide generated during the heating process. An eco-friendly pre-heater using infrared heating technology was developed for heating the pavement at a depth of 5 cm without burning the pavement surface. The new WIR equipment heats the existing asphalt pavement surface using an infrared heating burner with LPG fuel, which is designed to avoid a direct flame contact on the asphalt pavements. The temperature of the gas burner was over 1000 °C, and the

temperature at the pavement surface, including infrared conduction, was estimated at 800 °C. Figure 1 shows the new preheating equipment, second heater/paver, infrared heating unit, and temperature of the pavement surface after heating. The air temperature was measured as 35 °C and the temperature projected onto the pavement surface and the depth of 5 cm were 53 and 47 °C, respectively. As shown in Figure 1d, the equipment can heat the surface of pavement between 165 and 203 °C. Temperature sensors were installed to measure the temperature at 5 cm depth from the surface and the temperatures were measured between 70 to 73 °C. As a result, the warm surface milling was performed without crushing the aggregates.

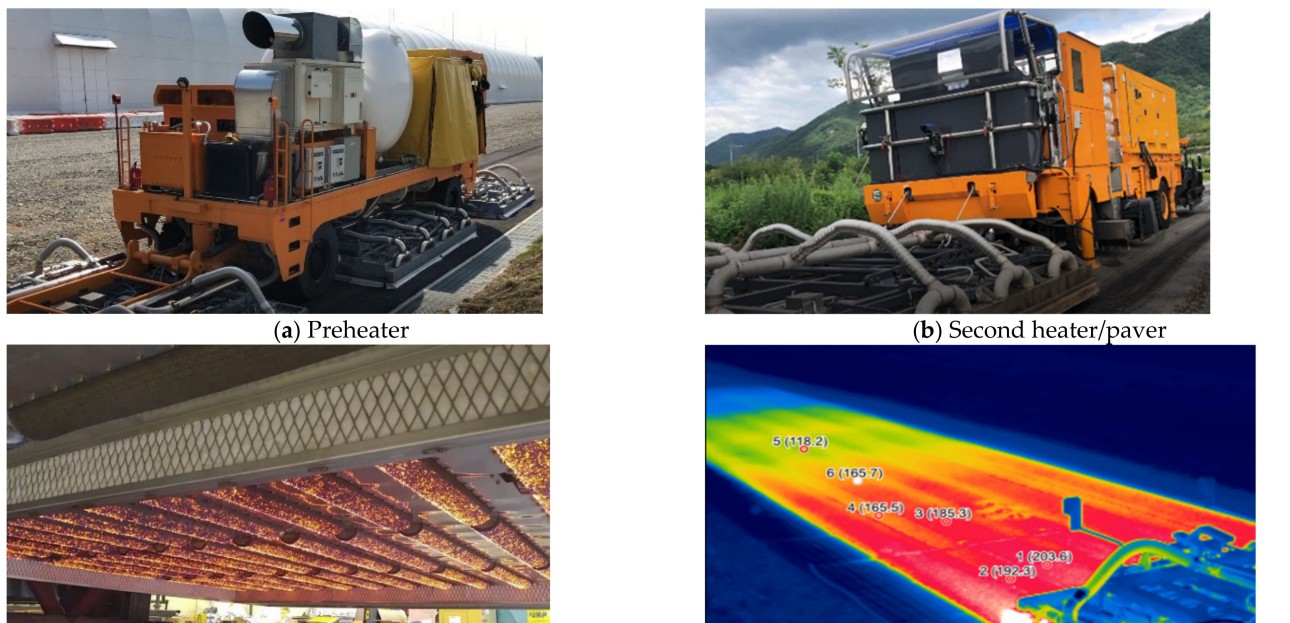

(**a**) Preheater

(**b**) Second heater/paver

(**c**) Infrared heating unit

(**d**) Temperatures of the heated pavement surface

**Figure 1.** New WIR equipment and temperature of pavement surface after heating.

As shown in Figure 2, an emission hood unit was installed on top of the heating plates where the collected emission is exhausted through the emission collector module. Figure 2a shows the emission control unit "off" condition and Figure 2b shows the "on" condition, where no emission is released. The emission-controlled heating equipment reduces the emission by capturing the VOCs generated during the heating process, which would lead to the eco-friendly pavement recycling practices.

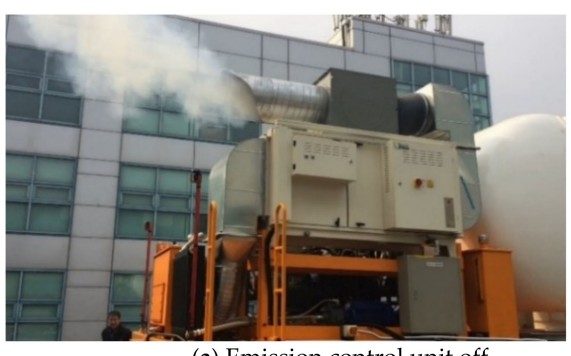
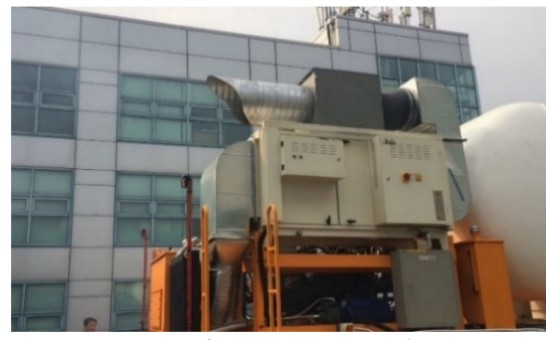

(**a**) Emission control unit off

(**b**) Emission control unit on

**Figure 2.** Heating equipment with the emission hood unit.

### 3. Warm In-Place Recycling Additives

To produce WIR additives, soybean oil was blended with various amounts of three commercially available amide type anti-stripping additives: (1) Ethylenediamine (ED); (2) 2-hydroxyethyl ethylenediamine (HEED); and (3) Tetraethylenepentamine (TEPA). ED ($C_2H_4(NH_2)_2$) is a colorless basic amine in liquid form, which readily reacts with moisture. HEED is a derivative of ED, which is a commercially available chelating agent. TEPA is another derivative of ED, which is used as a curing agent for epoxy. The formation of amide bonds was successfully achieved in the laboratory, which has been confirmed using the Fourier Transform Infrared (FTIR) spectroscopy and 1H-Nuclear Magnetic Resonance (NMR) equipment. The PG 64-22 binder with an antistripping additive synthesized from TEPA exhibited the higher moisture resistance value than those synthesized from ED and HEED. This paper evaluates the synergistic effect of TEPA with Soybean oil and Styrene-Butadiene-Styrene (SBS) polymer by applying 1) TEPA + Soybean Oil (A1) and 2) TEPA + SBS + Soybean oil (A2) on the Rolling Thin Film Oven (RTFO)-aged binder. To evaluate the short-term aging effect of these additives, the RTFO aged binder was adopted rather than the PAV-aged binder. The RTFO binders are conditioned in the oven at 163 °C for 85 min following the AASHTO T 24. To determine the optimum dosage rate of these WIR additives of A1 and A2, the viscosity and dynamic shear modulus were measured from RTFO-aged binders with various amounts of each additive.

### 3.1. Viscosity Test

Viscosity was measured using a rotational viscometer from 125 to 165 °C at 10 °C increments to better understand the behavior of asphalt in a wide range of temperatures. Figure 3 shows the viscosities of the RTFO-aged binders with three different dosages (5%, 10%, and 15%) of each additive and virgin PG 64-22 binder as a reference. At all temperatures, as shown in Figure 3, both additives (A1 and A2) lowered the viscosity of RTFO-aged binder to that of virgin binder and below as the dosage is increased from 5% to 15%. Overall, additive A1 consistently reduced the viscosity of the RTFO-aged binder more than additive A2 with SBS. The optimum dosages of A1 and A2 to lower the viscosity of RTFO-aged binder to that of virgin binder were found to be 3.7% and 11%, respectively.

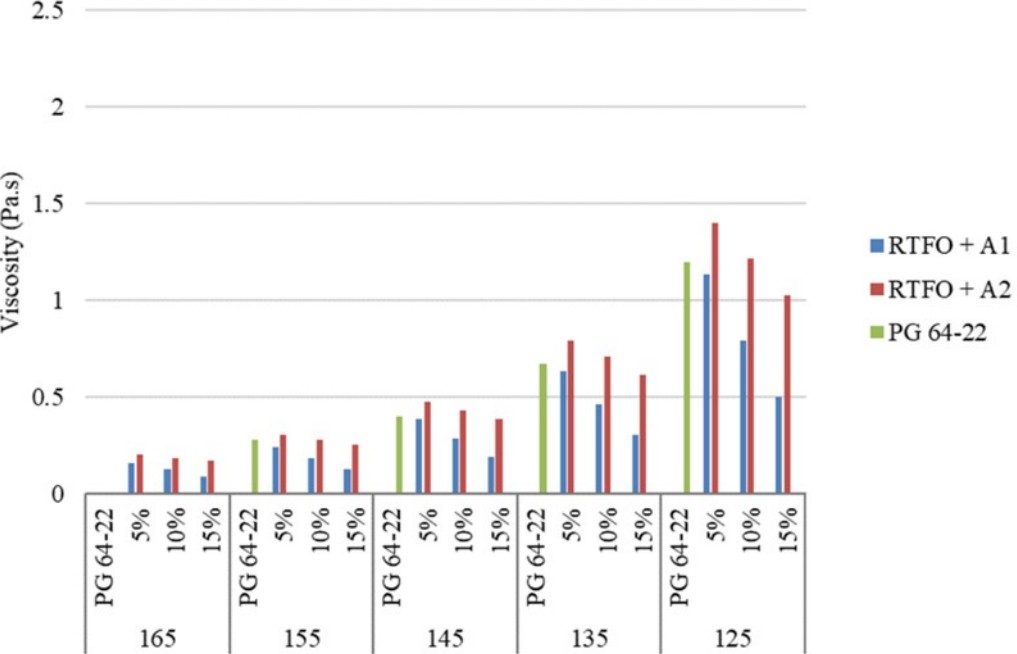

**Figure 3.** Viscosity of the RTFO-aged samples with the additives of A1 and A2.

*3.2. Dynamic Shear Rheometer Test*

The Dynamic Shear Rheometer (DSR) test was conducted at temperatures from 46 to 76 °C at 6 °C increments. Figure 4 shows the dynamic shear moduli of the RTFO-aged binders with three different dosages (5%, 10%, and 15%) of each additive and virgin PG 64-22 binder as a reference. At all temperatures, as shown in Figure 4, both additives (A1 and A2) lowered the dynamic shear modulus of RTFO-aged binder below that of virgin binder when the dosages were less than 5%. Overall, both additives of A1 and A2 had a similar effect on the dynamic shear modulus. The optimum dosages of A1 and A2 to lower the dynamic shear modulus of RTFO to that of virgin binder were found to be 2.25% and 3.4%, respectively.

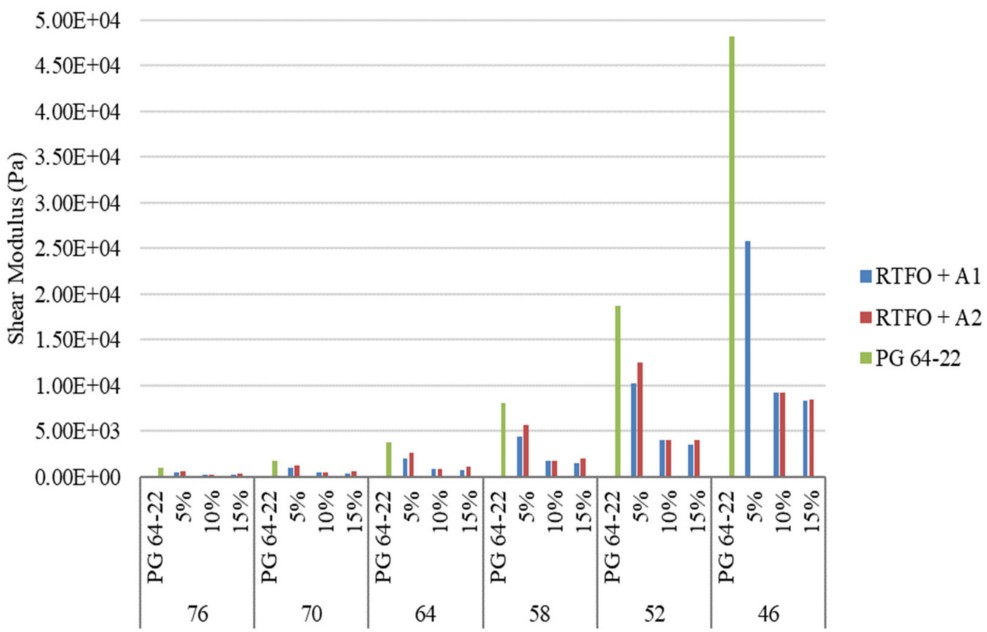

**Figure 4.** Dynamic modulus of the RTFO aged samples with the additives of A1 and A2.

## 4. WIR Mixture Performance Test

To identify the effect of two WIR additives (A1 and A2) on the performance of the 100% RAP mixtures, as shown in Figure 5a, the Hamburg Wheel Track (HWT) test was performed to determine their impacts on the moisture susceptibility, and in Figure 5b, the Disc-Shaped Compact Tension (DCT) test was performed to evaluate the low-temperature cracking potential.

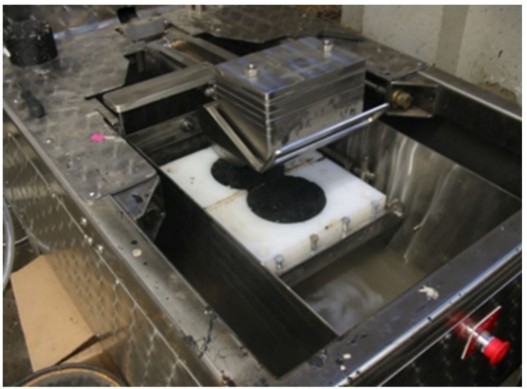

(**a**) Hamburg wheel tracking (HWT) test

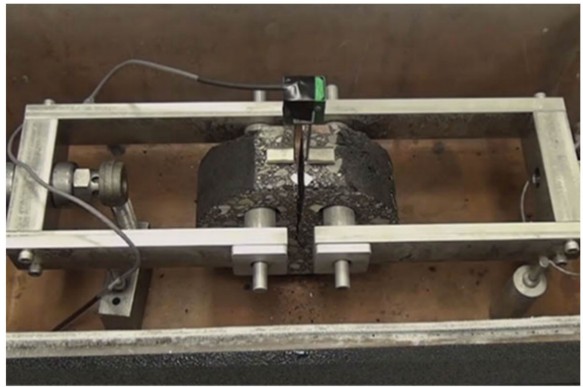

(**b**) Disc-shaped compact tension (DCT) test

**Figure 5.** Pictures of HWT and DCT testing equipment to determine moisture susceptibility and low-temperature cracking potential.

### 4.1. Aggregate Gradation

To determine the aggregate gradation of the RAP materials and asphalt contents, five RAP samples were burned off using the burn-off oven. The average gradation of five samples is plotted in Figure 6. Although the aggregate gradation did not meet the Superpave gradation requirements, it was adopted as is for this research since the purpose of the study is to use the in-place RAP materials as is without adding additional aggregates in the field.

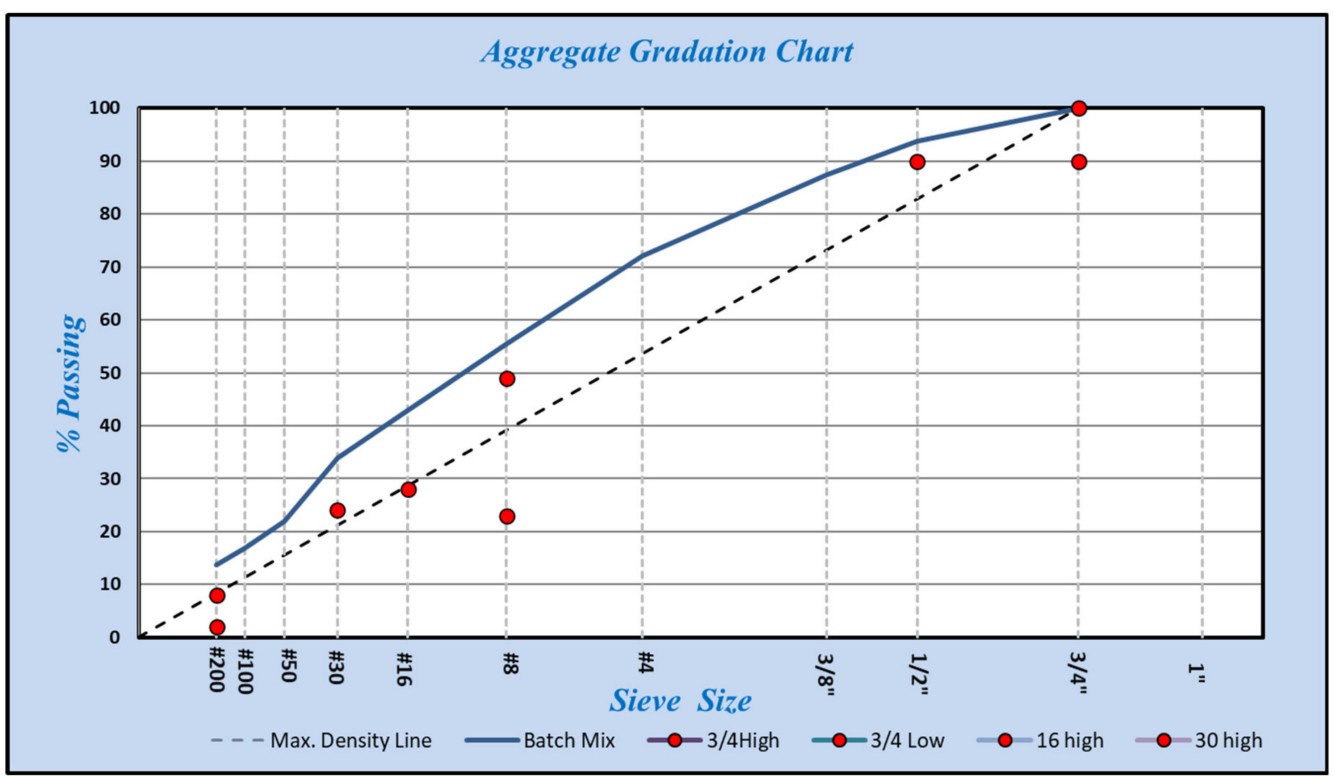

**Figure 6.** Gradation of extracted aggregates from the RAP stockpile.

### 4.2. RAP Binder Content

The mixing and compaction temperatures of the control HMA mixtures in the laboratory were selected as 155 and 145 °C and those of WIR mixtures were lowered by 20 °C to 135 and 125 °C, respectively. Since the 100% RAP materials were to be used, the bitumen content in RAP materials would play a key role in determining the optimum asphalt content. Since RAP materials were stockpiled outside, they contained high moisture contents. To determine the moisture content, they were dried in the oven at 120 °C for 2 h. The reason for applying the comparatively lower temperature and short drying time is to avoid asphalt burning of the binder in RAP. Next, to determine the asphalt content, the weights of five samples before and after the burn-off test were measured. The average asphalt and moisture contents of the five samples were 4.86% and 4.79%, respectively.

As summarized in Table 1, the Superpave mix design was performed on the 100% RAP materials and the air voids were below the optimum value of 4%. To increase the air voids to 4%, the asphalt binder should have been reduced or a new aggregates structure should have been needed. Since neither of them can be done for in-place recycling, samples with 100% RAP materials were compacted as is.

**Table 1.** Mix design properties of 4.86% bitumen content existing in RAP materials.

| Mix Design Properties | Value | Mix Design Criteria |
|---|---|---|
| Air Voids | 1.76% | 4.00% |
| Binder Content (%) | 4.86% | n.a. |
| Bitumen Absorption Pba | 0.38% | n.a. |
| Effective Bitumen Content Pbe | 4.44% | n.a. |
| Voids in Mineral Aggregate (VMA, %) | 12.32% | Minimum 14% |
| Voids Filled with Asphalt (VFA, %) | 84.87% | 75–85% |
| Dust to Binder Ratio (P0.075/Pbe) | 3.1 | 0.6–1.4 |
| Film Thickness (μm) | 14.31 | 8.0–15 μm |
| Gmb | 2.412 | n.a. |
| Gmm | 2.458 | n.a. |

*4.3. Hamburg Wheel Track Testing of 100% RAP Mixtures*

The Hamburg Wheel Track test was performed to evaluate the moisture susceptibility of 100% RAP mixtures with and without WIR additives. As summarized in Table 2, three samples were prepared for each of five different mixture types of the control HMA mix without additive and HMA and WIR mixes with the additives of A1 and A2.

**Table 2.** Mixing and compaction temperatures for each mix type.

| No. | Material | Type of Additive | Mixing Temperature | Compaction Temperature |
|---|---|---|---|---|
| 1 | 100% RAP | HMA without additive | 155 °C | 145 °C |
| 2 | 100% RAP | HMA with A1 | 155 °C | 145 °C |
| 3 | 100% RAP | HMA with A2 | 155 °C | 145 °C |
| 4 | 100% RAP | WIR with A1 | 135 °C | 125 °C |
| 5 | 100% RAP | WIR with A2 | 135 °C | 125 °C |

To evaluate the effect of additives on moisture susceptibility of 100% RAP mixtures, the dosage rate of the additives was fixed as 10% of the binder (0.486% by weight of the mix) for all samples. A dosage rate of 10% was selected to match the optimum dosage of 11% for A2 and the same dosage rate was used for A1 although its optimum dosage was 3.7%. We chose the same dosage rate for both A1 and A2 to compare their relative moisture susceptibility. For each mix type, three samples (two specimens are needed for each sample) were prepared with a target air void of 7.0% ± 0.5%. Table 3 summarizes the number of gyrations applied for each specimen to get 7% air voids. It is interesting to note that WIR with A1 required the least number of gyrations of 13.3 followed by HMA with A2. The test results for all the samples are summarized in Table 4 and the average value of three samples for each mix type is plotted in Figure 7. As can be seen from Figure 7, HMA without additive performed best followed by HMA with A2, WIR with A2, HMA with A1, and WIR with A1. Overall, mixtures with A2 with SBS performed better in HWT tests than mixtures with A1.

**Table 3.** Number of gyrations applied for compacting HWT samples at 7% air voids.

| Group | Material | Mix | Samples | | | | | | Average |
|---|---|---|---|---|---|---|---|---|---|
| | | | 1 | 2 | 3 | 4 | 5 | 6 | |
| 1 | RAP | HMA | 21.5 | 18 | 19 | 22 | 26 | 26 | 21.5 |
| 2 | RAP+A1 | HMA | 21.5 | 19 | 18 | 24 | 22 | 24 | 21.5 |
| 3 | RAP+A2 | HMA | 16 | 16 | 14 | 18 | 15 | 15 | 15.7 |
| 4 | RAP+A1 | WMA | 14 | 14 | 14 | 12 | 15 | 11 | 13.3 |
| 5 | RAP+A2 | WMA | 28 | 22 | 22 | 21 | 26 | 20 | 23.2 |

**Table 4.** Hamburg wheel test results for all the samples of all mixture groups.

| Mix Type | Test ID | Air Voids, % | Total Number of Passes | Slope Inflection Point | Max. Rut Depth, mm |
|---|---|---|---|---|---|
| RAP @HMA Temp | RAP 1 | 6.48% | 20,000 | >20,000 | 1.7 |
| | RAP 2 | 6.95% | 20,000 | >20,000 | 2.1 |
| | RAP 3 | 6.70% | 20,000 | >20,000 | 1.8 |
| | Average | 6.71% | 20,000 | >20,000 | 1.9 |
| RAP+A1@HMA Temp | RAP+A1-1 | 7.10% | 20,000 | 10,501 | 4.3 |
| | RAP+A1-2 | 7.15% | 20,000 | 9397 | 10.6 |
| | RAP+A1-3 | 6.56% | 20,000 | 10,243 | 7.0 |
| | Average | 6.86% | 20,000 | 10,086 | 7 |
| RAP+A2 @HMA Temp | RAP+A2-1 | 6.56% | 20,000 | >20,000 | 3.2 |
| | RAP+A2-2 | 6.75% | 20,000 | >20,000 | 4.7 |
| | RAP+A2-3 | 6.86% | 20,000 | >20,000 | 3.0 |
| | Average | 6.72% | 20,000 | >20,000 | 3.6 |
| RAP+A1 @WMA Temp | RAP+A1-1 | 6.87% | 20,000 | 9456 | 16.3 |
| | RAP+A1-2 | 7.33% | 15,800 | 9238 | 20.0 |
| | RAP+A1-3 | 7.15% | 14,400 | 2221 | 20.0 |
| | Average | 7.24% | 16,733 | 5025 | 19 |
| RAP+A2 @WMA Temp | RAP+A2-1 | 6.85% | 20,000 | 13,235 | 5.1 |
| | RAP+A2-2 | 6.98% | 20,000 | 13,357 | 5.3 |
| | RAP+A2-3 | 6.74% | 20,000 | 13,425 | 5.7 |
| | Average | 6.86% | 20,000 | 13,497 | 5 |

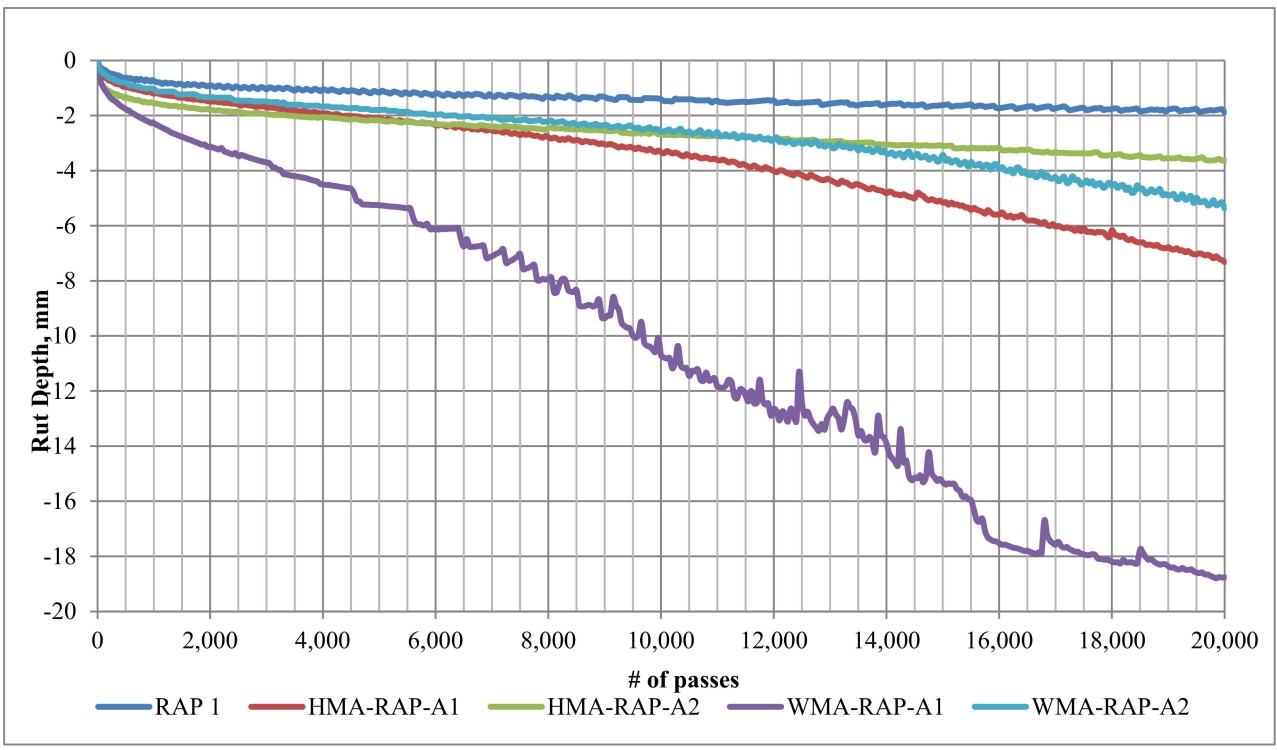

**Figure 7.** Average test results of the Hamburg wheel tracking test for all mixture types.

### 4.4. Disk-Shaped Compact Tension Test

The Disc-Shaped Compact Tension (DCT) test was performed to evaluate the low-temperature cracking potential of the following 100% RAP mixtures: (1) WIR without additive; (2) WIR with A1; and (3) WIR with A2. The minimum dosage for A1 was determined as 3.7% during the viscosity test. However, we increased it to 7% while keeping

the dosage of A2 as 11%, so that a similar amount of soybean oil will be present in both additives. Three WIR samples were mixed at 125 °C and compacted at 115 °C. A standard test temperature for DCT specimens was selected as –12 °C, which is 10 °C warmer than the PG low temperature limit of –22 °C. As shown in Figure 8, loads applied on three samples are plotted against Crack Mouth Opening Displacement (CMOD). As can be seen from this figure, the RAP mix with A2 exhibited not only a higher peak load but also a higher fracture energy (calculated as an area under the curve). The RAP mix with A1 exhibited a slightly lower peak load but its peak load did not drop as quickly resulting in a higher fracture energy than the RAP mix without additive.

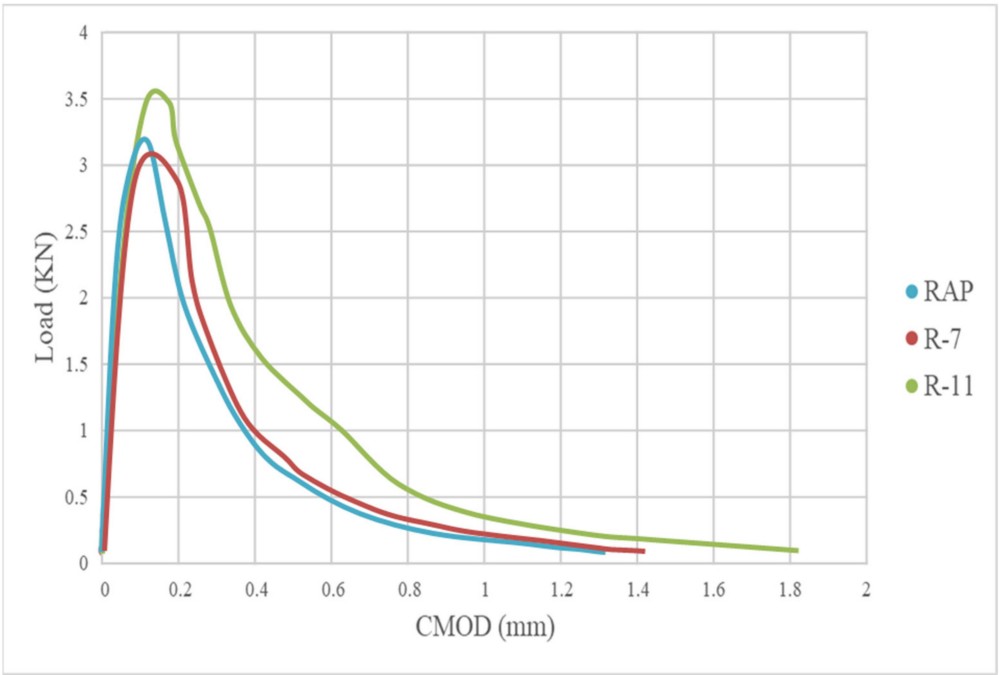

**Figure 8.** Load vs. CMOD of RAP specimens without additive, 7% A1 and 11% A2.

Fracture energy values for all samples are summarized in Table 5 and the average fracture energy value for each mix type is plotted in Figure 9. A fracture resistance of 100% RAP mixtures is increased as A2 or A1 is added. It can be concluded that A2 is more effective in increasing the fracture resistance than A1.

**Table 5.** Fracture energy (J/m) of all the specimens.

| Mix Type | Fracture Energy (J/m) | | | | |
|---|---|---|---|---|---|
| | Sample #1 | Sample #2 | Sample #3 | Sample #4 | Average |
| 100% RAP | 222 | 247 | 257 | 213 | 234.75 |
| 100% RAP + 7% A1 | 367 | 294 | 361 | – | 340.6 |
| 100% RAP + 11% A2 | 391 | 345 | 426 | 389 | 387.5 |

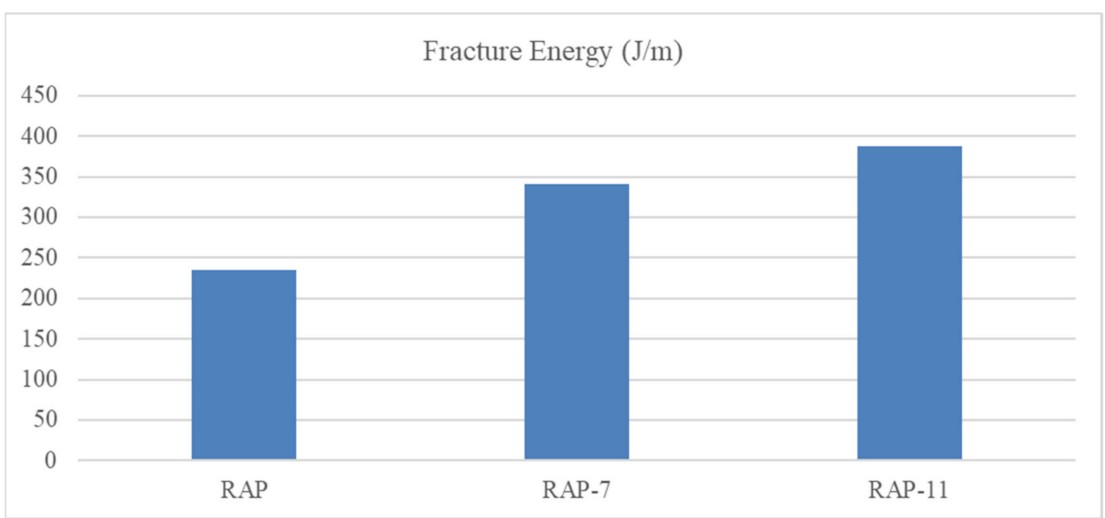

**Figure 9.** Average fracture energy of specimens.

## 5. Summary and Conclusions

This paper discusses a new Warm In-place Recycling (WIR) technology that can overcome the limitations of HIR by lowering a heating temperature of HIR while adding a warm mix asphalt additive. This paper evaluates two WIR additives of A1 (Tetraethylenepentamine (TEPA) + soybean oil) and A2 (TEPA + soybean oil + SBS polymer). To determine the optimum dosage rate of these WIR additives of A1 and A2, both viscosity and dynamic shear modulus were measured from RTFO-aged binders with various amounts of each additive. The optimum dosages of A1 and A2 to lower the viscosity and dynamic modulus of RTFO-aged binder to that of virgin binder were found to be 3.7% and 11%.

To identify the effect of two WIR additives (A1 and A2) on moisture sensitivity and low-temperature cracking potential of the 100% RAP mixtures, the Hamburg Wheel Track (HWT) and Disc-Shaped Compact Tension (DCT) tests were performed, respectively. To ensure an amount of TEPA and soybean oil equivalent to that of A2, a dosage rate for the additive A1 was increased to 7% while keeping an optimum dosage rate of A2 as 11%.

To evaluate moisture sensitivity, HWT tests were performed three times on each of five different mix types of two WMA mixes with A1 or A2 (mixing at 155 °C and compaction at 145 °C), two HMA mixes with A1 or A2 and one HMA mix without additive as a control (mixing at 135 °C and compaction at 125 °C). Based on HWT test results, HMA without additive performed best followed by HMA with A2, WIR with A2, HMA with A1, and WIR with A1. Overall, mixtures with A2 with SBS performed better in HWT tests than mixtures with A1 without SBS.

To evaluate the low-temperature cracking potential, DCT tests were performed on each of three WMA mixes (mixed at 125 °C and compacted at 115 °C). A fracture resistance of 100% RAP mixture increased as either A2 or A1 is added where A2 with SBS was more effective in increasing the fracture resistance than A1 without SBS.

Finally, a new WIR equipment was developed for recycling asphalt pavements with significantly reduced emission amounts. The infrared heating equipment was designed to heat the pavement surface at between 165 and 203 °C and, based on the temperature sensors embedded at the depth of 5 cm from the pavement surface, temperatures ranged between 70 and 73 °C. WIR additives were sprayed when scraping the pavement. The additives were mixed using a newly designed two-axis continuous mixer and were mixed for less than 1 min.

Based on the laboratory tests and field trials, the following conclusions can be made:

- The WIR additive was successfully produced by forming an amide bond among ethylenediamines, soybean oil, and SBS polymer and applied in the field trial.

- Based on laboratory HWT and DCT tests, the WIR additive with TEPA, soybean oil, and SBS polymer performed better than the WIR additive without SBS.
- The emission-controlled heating equipment was successfully manufactured using indirect infrared heating equipment to reduce the emission by capturing the volatile organic compounds generated during the heating process, which would lead to the eco-friendly pavement recycling practices.

**Author Contributions:** Conceptualization, H.L. and S.-A.K.; methodology, N.-J.C., formal analysis, B.M. and A.B.; data curation, A.B. and B.M.; invesgtation, K.-D.J. and N.-J.C.; project administration, S.-A.K. and N.-J.C.; writing—original draft preparation, B.M. and A.B.; writing—review and editing, H.L. All authors have read and agreed to the published version of the manuscript.

**Funding:** This research received funding from the Korea Agency for Infrastructure Technology Advancement (KAIA).

**Institutional Review Board Statement:** Not applicable.

**Informed Consent Statement:** Not applicable.

**Conflicts of Interest:** The authors declare no conflict of interest.

**Disclaimer Notice:** The contents of this paper reflect the views of the authors, who are responsible for the facts and the accuracy of the information presented herein. The opinions, findings, and conclusions expressed in this paper are those of the authors and not necessarily those of the sponsors. The sponsors assume no liability for the contents or use of the information contained in this document.

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
