# Peer review of "Development of Warm In-Place Recycling Technique as an Eco-Friendly Asphalt Rehabilitation Method"

_infrastructures, doi:10.3390/infrastructures6070101_

Round 1
Reviewer 1 Report
The paper "Development of Warm In-Place Recycling Technique as An Eco-Friendly Asphalt Rehabilitation Method" is suitable for publication in Infrastructure journal, after necessary corrections:
a) In the abstract there are loose elements and no information, for example, "New WIR equipment was developed for recycling asphalt pavements with significantly reduced amounts of emission" What reduction is this? I think you should mention more quantitative details in this part of your role;
b) The literature covered is very limited, and in some cases old. I suggest taking an initial approach on the sustainability of new construction materials, consider these roles: 10.1016 / j.cscm.2020.e00406; 10.1016 / j.scitotenv.2021.146448; 10.1016 / j.jclepro.2021.127225. Another interesting topic is a greater contextualization of asphalt pavements, note these roles: 10.1016 / j.cscm.2021.e00556; 10.1520 / JTE20200465; 10.1007 / 978-3-030-65493-1_59; 10.1520 / JTE20200276. Consider these all papers, and others that you deem necessary, your final paper has 19 articles, it should be more complete in that sense. Many parts of your text are not even cited;
c) "To identify the effect of two WIR additives (A1 and A2) on performance of the 100% RAP mixtures, 1) Hamburg Wheel Track (HWT) was performed to determine their impacts on the moisture susceptibility, and 2) the Disc- Shaped Compact Tension (DCT) test was performed to evaluate the low-temperature cracking potential.The dosage rates of A1 and A2 were determined as 7% and 11% based on the viscosity and dynamic shear modulus test results. to ensure an amount equivalent to that of A2, a dosage rate for the additive A1 was nearly doubled from the minimum dosage rate of 3.7% based on the viscosity test. " All this information is not clear, moreover, they are not related to how these tests will be done, are they normative ??
d) "To evaluate the effect of additives on moisture susceptibility of 100% RAP mixtures, the dosage rate of the additives was fixed as 10% of the binder (0.486% by weight of the mix) for all samples." Why? You need to justify it.
e) "For each mix type, three samples were prepared with a target air voids of 7.0% ± 0.5%. To prepare three HWT samples, six specimens were prepared for each mix type. Table 3 summarizes the number of gyrations applied for each mix type to get 7% air voids. I "Is the number of samples you report confusing, based on what? It has statistical study !!!
d) In general, I note again that your role is poor in discussions and comparisons with other works in the literature, you do not do this, proof of this is not to have references in the discussion of your results, this must be strongly corrected for this role be accepted for publication !!!
Reviewer 2 Report
Congratulations. It is an interesting article about the possibilities of new asphalt pavement recycling techniques. Proposed Warm In-place Recycling (WIR) solution allows to reduce the emission of harmful compounds into the atmosphere, while obtaining good properties of the pavement layer. However, certain elements require clarification and respond in the paper. This mainly concerns the method of adding the WIR additive at the recycling stage and the effectiveness of this additive when dosing it to the RAP. Authors should refer to this in the article. Detailed comments are provided below:
- In point 2 it is written: „… the equipment can heat the surface of pavement between 165°C and 203°C, which heats a top 5 cm of an existing asphalt pavement to 121°C on average.” Fig. 1 shows the surface temperature of heated layer. On what grounds was it found that the average temperature of a 5 cm thick layer was 121°C?
- Under what external conditions (air temperature, humidity, wind speed) and layer parameters (temperature, humidity, thermal conductivity, heat capacity) can the surface be heated to an average layer temperature of 121°C? This information is not included in the paper.
- The new WIR equipment heats the existing asphalt pavement surface using an infrared heating burner with LPG fuel. The combustion temperature of LPG gas is approx. 700°C and with additional forced air flow it can even be as high as 1400°C. What the authors mean by "infrared burner". What is the temperature of the surface heating factor?
- In point 3 the results of PG 64-22 bitumen tests before and after RTFO aging are shown. Shouldn't bitumen also be exposed to long-term aging of PAV (especially in the case of recycling of wearing courses)?
- In point 3, the influence of WIR additives on the properties of the PG 64-22 base asphalt is analyzed. In real conditions the additives will be dosed into the RAP, in which the asphalt is in thin surface layer and hardened (after short-term and long-term aging). The problem may be the combination of the WIR additive with bitumen at lower temperatures (average approx. 120°C). Shouldn't the tests be carried out on asphalt extracted from RAP without additives and after mixing the RAP with additives?
- In point 4 the content of additives A1 and A2 was determined on the basis of the viscosity test and the Dynamic Shear Rheometer. The content of the A1 additive was increased almost twice (from 3.7% to 7%). The justification for increasing the content of this additives is unclear.
- At what stage on a technical scale and how will the WIR additive be dosed to RAP? WIR additives (A1 and A2) - these are ammonium compounds and are susceptible to elevated temperatures (limitation of their effectiveness as a result of high temperatures, above 165°C). How long is the mixing time of the WIR additive with RAP?
Reviewer 3 Report
The paper looks more like a technical note rather than a scientific article. The language and flow should be revised. Additionally, there are no line numbers, so it is hard to give comments.
- The abstract is not bad, but again it needs better flow.
- Page 1: The first paragraph of the introduction is not very precise. Like "the use of RAP increased by 8.5 percent in 2019 with an average of 21.1 percent." Where? In the world, in Africa, in Canada?
- It is an international journal, so not everybody knows what Ohio or Maryland is. When you talk about states, you should use US states. This comment is valid for the whole manuscript.
- "less hauling distance" I would say no hauling distance since it is in-place recycling.
- You need a literature review. Now you basically have some limited introduction.
- You don`t have objective(s). Please state clearly what you want to achieve.
- You should describe the difference between your equipment and common HIR equipment. I know that HIR uses a direct flame.
- Section 3 is vague. You need more details about your materials.
- I am guessing that you aged some binder to simulate field binder. You haven't extracted a binder from RAP to get test material?
- How have you obtained the rap?
- Section 4.2: It is hard to understand what you have done. Why do you have control HMA mixture? Do you compare your results to HMA? Have you compared it to CIR or HIR?
- What is the purpose of Table 3?
- Table 4: I think you don`t need this table. Better use figure.
- More than low-temperature cracking, I would be interested to see intermediate temperature fatigue cracking.
- Figure 7: The lines are too thick.
- Table 5: It is hard to read
- The summary and Conclusion section is just a summary. You need some general conclusions.
I think in the current form, the manuscript should not be accepted for publication unless it is submitted as a technical note.
Round 2
Reviewer 1 Report
The authors made the necessary corrections for the acceptance of the work.
Reviewer 3 Report
Thank you for addressing my comments.